# ALIGNFLOW: IMPROVING FLOW-BASED GENERATIVE MODELS WITH SEMI-DISCRETE OPTIMAL TRANSPORT

**Lingkai Kong**[1]*, **Molei Tao**[1],
**Yang Liu**[2], **Bryan Wang**[2], **Jinmiao Fu**[2], **Chien-Chih Wang**[2], **Huidong Liu**[2]
[1] Georgia Institute of Technology,   [2] Amazon.com, Inc.
{lkong75,mtao}@gatech.edu
{yliuu,brywan,jinmiaof,ccwang,liuhuido}@amazon.com

## ABSTRACT

Flow-based Generative Models (FGMs) effectively transform noise into complex data distributions. Incorporating Optimal Transport (OT) to couple noise and data during FGM training has been shown to improve the straightness of flow trajectories, enabling more effective inference. However, existing OT-based methods estimate the OT plan using (mini-)batches of sampled noise and data points, which limits their scalability to large and high-dimensional datasets in FGMs. This paper introduces AlignFlow, a novel approach that leverages Semi-Discrete Optimal Transport (SDOT) to enhance the training of FGMs by establishing an explicit, optimal alignment between noise distribution and data points with guaranteed convergence. SDOT computes a transport map by partitioning the noise space into Laguerre cells, each mapped to a corresponding data point. During FGM training, i.i.d. noise samples are paired with data points via the SDOT map. AlignFlow scales well to large datasets and model architectures with negligible computational overhead. Experimental results show that AlignFlow improves the performance of a wide range of state-of-the-art FGM algorithms and can be integrated as a plug-and-play component. Code is available at: https://github.com/konglk1203/AlignFlow.

## 1 INTRODUCTION

A generative model in machine learning is designed to produce new data samples that closely resemble those drawn from a given dataset. This task is of fundamental importance and has seen significant advances over the past decades. Notable examples include autoregressive models such as GPT (Radford et al., 2018) and ChatGPT (Achiam et al., 2023) for natural language generation, and diffusion models (Sohl-Dickstein et al., 2015; Ho et al., 2020; Song et al., 2021) for image synthesis (e.g., Rombach et al., 2022). In addition, other prominent generative modeling approaches include Generative Adversarial Networks (GANs) (Goodfellow et al., 2020), Normalizing Flows (Rezende & Mohamed, 2015), Rectified Flow (Liu et al., 2022), Flow Matching (Lipman et al., 2022), and Stochastic Interpolant (Albergo et al., 2023).

This work focuses on improving a wide range of Flow-based Generative Models (abbreviated as FGMs hereafter), including Flow Matching (Lipman et al., 2022), shortcut model (Frans et al., 2025), MeanFlow (Geng et al., 2025), Live Reflow (Frans et al., 2025), but excluding continuous normalizing flows (CNF) (Chen et al., 2018); see Sec. 3.1 for a detailed specification of scope. FGMs focus on learning a time-dependent vector field, approximated by a Neural Network (NN) whose integration gives a trajectory that transforms a randomly sampled noise to a generated datum.

Despite their ability to generate high-quality samples, FGMs (like other related methods such as diffusion models) can require substantial computational cost in their inference step (i.e. sampling), where an Ordinary Differential Equation (ODE) is numerically integrated and each integration step involves at least one forward pass of a large NN. Consequently, generating a single sample requires multiple NN evaluations. This cost is typically measured by the Number of Function Evaluations

---

*This work was primarily conducted during Lingkai Kong's internship at Amazon.

(NFEs), which denotes the total number of forward passes during ODE integration. For example, in vanilla Flow Matching (Lipman et al., 2022), the NFEs are often greater than 100. The straightness of the ODE trajectory is closely related to NFEs, as a straighter trajectory is typically easier to integrate, resulting in fewer NFEs required. See Apdx. D.2 for the inference procedure of FGMs.

Before diving into how to improve trajectory straightness, we first review the training procedure of FGMs. Each training iteration of FGMs generally involves three main steps: (1) sampling noise and data points; (2) constructing the target vector field that a NN is trained to approximate; and (3) updating the model parameters. For implementation details, see Algorithms 1 and 4. In this work, we focus on improving the first step. While many State-of-the-Art (SOTA) methods introduce various target vector fields to promote straighter flow trajectories in step 2, noise and data points are still typically sampled independently in step 1. However, such independence has been shown to inherently induce curved flow trajectories and lead to high NFEs during sampling. In other words, the random matching of noise and data points implicitly encourages non-straight generative paths, motivating carefully designed couplings (e.g., Liu et al., 2022; Hertrich et al., 2025).

One family of approaches for making the trajectories straighter is based on using Optimal Transport (OT) to couple noise and data more effectively (Tong et al., 2023; Pooladian et al., 2023; Kornilov et al., 2024; Cheng & Schwing, 2025). From a theoretical perspective, OT gives a shortest path between the noise and data distributions (Eq. (4)), yielding inherently straight mappings suitable for coupling. Despite the appealing theoretical properties of OT, scaling OT-based FGMs to large-scale problems remains challenging (Sec. 2). Utilizing discrete OT, Tong et al. (2023) and Pooladian et al. (2023) couple the noise samples and data samples within each minibatch. However, discrete OT plans between minibatches of samples are known to provide biased (Kornilov et al., 2024) or misspecified (Nguyen et al., 2022) approximation due to a fundamental challenge, namely that an accurate estimation of the OT plan in this setting requires the number of noise samples to grow exponentially with the data dimensionality, known as the curse of dimensionality (Dudley, 1969; Genevay et al., 2019). In contrast to discrete OT-based methods, Kornilov et al. (2024) uses a continuous OT approach by representing the OT map using a Brenier potential parametrized by an Input Convex Neural Network (ICNN) (Amos et al., 2017). Their algorithm jointly optimizes the transport map and the Flow Inversion Map. Nonetheless, the optimality and convergence of the learned transport map are not established.

We therefore propose a different approach, AlignFlow, based on the following observation: to train a FGM, there are only finite amount of training data, corresponding to a discrete empirical distribution, but we can (and typically do) use a noise distribution that is continuous. Therefore, we seek to optimally couple discrete distribution with continuous distribution, and the OT problem corresponding to this case is known as Semi-Discrete Optimal Transport (SDOT). This is different from the aforementioned discrete OT approaches that use both data and noise samples to estimate an OT map or the continuous OT approach that introduces extra learning component with inductive bias (ICNN). Nevertheless, as a special case of OT, SDOT still inherits its desirable theoretical properties: it defines the most direct and shortest connection between noise samples and data points, thereby guiding the target vector field learned by a NN. Moreover, the SDOT map can be solved efficiently with guaranteed convergence, and the quality of the resulting map can be evaluated with low computational cost (Liu et al., 2021). In addition, SDOT also has a clear geometric interpretation: the SDOT map partitions the space into Laguerre cells (Fig. 1), where each cell corresponds to a data point, and all noise samples residing in that cell are optimally aligned to the data point.

SDOT map is a deterministic map from noise to data, unlike a general coupling which can match one noise sample to different data probabilistically (or vice versa). Such determinism is a necessary condition for achieving optimal coupling and ensuring consistent noise-data matching regardless of batch size (Sec. 4) in training FGMs. Motivated by the favorable properties of the determinism of SDOT, we call special couplings that are given by a deterministic mapping from the noise space to a dataset Noise–Data Alignment (NDA) (Def. 1). While many prior works leverage OT to improve FGMs, the key innovation of AlignFlow is its use of SDOT to explicitly construct an NDA.

The training procedure of AlignFlow follows a two-stage approach: in the first stage, the SDOT map is computed. In the second stage, a general FGM is trained using this precomputed SDOT map. The key advantages for AlignFlow are summarized below:

- AlignFlow is a *plug-and-play* method that can be seamlessly integrated into existing FGMs. It readily combines with SOTA FGMs to further enhance performance.

- AlignFlow has its NDA component separated from FGM training, and therefore enjoys the provable convergence of SDOT in (Liu et al., 2021). In contrast, previous OT-based FGMs lack convergence guarantees for the computation of coupling between noise and data.

- AlignFlow *bypasses the curse of dimensionality* (Sec. 4.1) that arises when using discrete OT for continuous distributions (Genevay et al., 2019), enabling it to scale effectively to large models and high-dimensional datasets.

- The SDOT map defines a deterministic and optimal transport path without randomness; that is, each noise sample is consistently matched to a fixed data point, independent of batch size. This batch-invariance ensures stable convergence (see Sec. 4.2) even when batch sizes are severely constrained, addressing a critical bottleneck for training large-scale models where memory limitations enforce small-batch regimes.

- The extra cost for computing the SDOT map is low (less than 1% extra cost) (Sec. 4.3).

## 2 RELATED WORK

A central factor in reducing the NFEs during sampling in FGMs is the straightness of the generative trajectory. Since the inference process involves integrating a learned vector field, straighter trajectories are easier to integrate accurately, thereby reducing NFEs. Numerous methods have been proposed to encourage straighter trajectories, which can be broadly categorized into three main approaches:

**Target vector field**: This class of methods aims to straighten generative trajectories by guiding the NN to learn a smoother or more linear target vector field. Approaches such as consistency models (Song et al., 2023) and shortcut models (Frans et al., 2025), inspired by techniques from diffusion models (Yang et al., 2024), enforce penalties on inconsistencies between the forward and backward segments of the trajectory. MeanFlow (Geng et al., 2025) introduces a stronger regularization term based on the Jacobian-vector product. These advanced loss functions have proven effective, reducing the NFEs to as low as 4 and even 1 in class-conditioned generation tasks on the ImageNet dataset.

**Distillation**: Recent works aim to reduce NFEs by distilling a trained FGM into a more efficient model (Boffi et al., 2025; Dao et al., 2025). Distillation techniques have also shown success in diffusion models (Salimans & Ho, 2022; Song et al., 2023; Kim et al., 2023). However, distillation involves an additional training stage built upon a pre-trained model, and thus can be viewed as complementary to our approach.

**Coupling**: Coupling is a joint distribution over two marginals. In standard FGMs, noise and data are independently paired during training. Several recent works aim to improve FGMs by introducing better coupling strategies by selecting noise and data from a carefully designed joint distribution:

- Tong et al. (2023) and Pooladian et al. (2023) employ the Sinkhorn algorithm (Peyré et al., 2019, Sec. 4.2) to compute the OT plan between i.i.d. sampled noise and data points in each FGM training iteration. However, this approach is sensitive to batch size: large batches incur high computational cost, as Sinkhorn is an iterative algorithm with per-batch complexity of $\mathcal{O}(B^2 \log B)$ where $B$ is the batch size, while small batch size leads to poor estimates of the optimal coupling. Calvo-Ordonez et al. (2025) introduce a weight from Gibbs kernel in the FGM training objective, which generalizes the idea of Tong et al. (2023). However, its dependence on minibatch makes it have similar weaknesses. Additionally, computing discrete OT in batches is known to produce biased transport plans (Kornilov et al., 2024), or misspecified mappings (Nguyen et al., 2022).

- Kornilov et al. (2024) train an Input Convex Neural Network (ICNN) (Amos et al., 2017) to serve as the Brenier potential (Peyré et al., 2019, Thm. 2.1), thereby providing the OT map between the noise and real data distributions. The ICNN and the Flow Inversion Map (Sec. C.a in Kornilov et al. (2024)) are optimized alternatively. However, the convergence property and optimality of the implicit OT plan approximated by ICNN remain unestablished.

- Zhang et al. (2025) provides an approach that scales the Sinkhorn algorithm to large-scale datasets, computing OT couplings between sampled noise and data points in large batches. Moreover, PCA can be used to further speed up Sinkhorn computation, which Zhang et al. (2025) reports does not sacrifice generation quality. Davtyan et al. (2025) improves upon

| | Algorithm | Scale to large models | Entire noise distribution | Explicit OT optimization |
|---|---|---|---|---|
| Coupling | Tong et al. (2023) | ✗ | ✗ | ✓ |
| | Pooladian et al. (2023) | ✗ | ✗ | ✓ |
| | Liu et al. (2022) | ✓ | ✗ | ✗ |
| | Kornilov et al. (2024) | ✗ | ✗ | ✗ |
| Noise-Data Alignment | **AlignFlow (ours)** | ✓ | ✓ | ✓ |

Table 1: A comparison between coupling methods and AlignFlow, a Noise-Data Alignment method.

mini-batch OT by computing the map between a pre-sampled noise set and the whole dataset. Nevertheless, it is limited by its ability to handle only a finite number of noise samples.

- Liu et al. (2022) propose to disentangle crossing trajectories in the learned vector field to promote straighter generative trajectories. This approach has recently been scaled to larger models by Esser et al. (2024), although the scaling process is non-trivial. While the method does not directly solve an OT problem, its underlying formulation is closely related to OT, as discussed in Theorem 3.5 of their paper.

A comparison is provided in Table 1. While conceptually related, AlignFlow differs fundamentally from existing coupling-based methods: it computes an SDOT map by considering the entire noise distribution and all empirical data points, resulting in optimal NDA for training FGMs.

*After ICLR decision, we saw the announcement of a nice concurrent work Mousavi-Hosseini et al. (2025). Both their work and this one demonstrate the effectiveness and robustness of the idea.*

## 3 METHODOLOGY

Mathematically, the generative modeling task can be formulated as follows: given a dataset $\{x_1^i\}_{i \in I}$, assumed to consist of i.i.d. samples from an unknown probability distribution $\tilde{p}_1$ on a space $\mathcal{X}$[1], where $I$ is the index set of the dataset [2]. The objective is to generate new samples that follow the same distribution $\tilde{p}_1$.

**Notation** In this paper, we establish the following notational conventions. We denote by $p_0$ the noise prior distribution (e.g., a standard Gaussian distribution) from which sampling is straightforward. Conversely, $p_1$ represents the empirical data distribution, defined as a Dirac mixture $p_1 = \sum_{i \in I} b_i \delta_{x_1^i}$. In our FGM training setup, the weights are uniform, i.e., $b_i = 1/|I|$.

### 3.1 FLOW-BASED GENERATIVE MODELS (FGMs)

We begin by summarizing a general FGM framework in Algo. 1. In each training iteration, a set of noise $x_0$, data $x_1$ and time $t$ are drawn. $x_t$ are computed as the interpolation between $x_0$ and $x_1$. Finally, a NN $u(x_t, t; \theta)$ is trained to approximate the target vector field TargetVectorField and the NN parameters $\theta$ are updated accordingly.

**Examples:** (Vanilla) Flow Matching (Lipman et al., 2022) defines the target vector field as $\text{TargetVectorField}(x_0, x_1) = x_1 - x_0$ and uses the loss function $\text{Loss}\left(\left\{\hat{v}^j, v^j\right\}_{j=1}^{B}\right) = \frac{1}{B}\sum_j \left\|\hat{v}^j - v^j\right\|_2^2$. Several existing methods, including the shortcut model (Frans et al., 2025), MeanFlow (Geng et al., 2025), Consistency Training (Frans et al., 2025), Live Reflow Frans et al. (2025), conform to this general FGM framework. See Sec. D.1 for more details.

---

[1]In many settings, $\mathcal{X}$ is chosen as a latent space, with data obtained via encoding through a Variational Autoencoder (VAE).

[2]Without loss of generality, we assume $I = \{1, 2, ..., N\}$, where $N$ is the dataset size.

---

**Algorithm 1:** Flow-based Generative Models (Training)

---

1   Input: Source distribution $p_0$, dataset $\{x_1^i\}_{i \in I}$, neural network $u(x, t; \theta)$
2   Output: Learned velocity field $u(\cdot, \cdot; \theta)$
3   Hyperparameters: batch size $B$, number of steps $K$
    ▷ Training loop
4   for $k = 1 \ldots K$ do
5       Sample i.i.d. $\{x_0^j\}_{j=1}^B \sim p_0$                         ▷ sample noise
6       Sample minibatch $\{x_1^{m_j}\}_{j=1}^B$ from dataset $\{x_1^i\}_{i \in I}$       ▷ sample data
7       $t^j \sim \mathcal{U}(0, 1)$ for $j = 1, \ldots, B$                   ▷ sample time
8       for $j = 1 \ldots B$ do
9           $x_t^j \leftarrow (1 - t^j)x_0^j + t^j x_1^{m_j}$
10         $v^j = \text{TargetVectorField}(x_0^j, x_1^{m_j})$
11         $\hat{v}^j = u(x_t^j, t^j; \theta)$
12       $L(\theta) = \text{Loss}\left(\{\hat{v}^j, v^j\}_{j=1}^B\right)$             ▷ objective function
13       Update $\theta$ using $\nabla_\theta L$                   ▷ optimization

---

In lines 5 and 6 of Algo. 1, noise and data points are sampled independently, and thus paired randomly, leading to curved trajectories intrinsically (Liu et al., 2022; Hertrich et al., 2025).

### 3.2   COUPLING BETWEEN NOISE AND DATA

In fact, the loss function in Algo. 1 provides an empirical estimate of the following expectation [3]:

$$L(\theta) \approx \mathcal{L}(\theta) = \mathbb{E}_{t \sim \text{Unif}[0,1], x_0 \sim p_0, x_1 \sim p_1} \|u(x_t, t; \theta) - \text{TargetVectorField}(x_0, x_1)\|_p^p \quad (1)$$

This formulation implies that $x_0$ and $x_1$ are sampled independently from $p_0$ and $p_1$, respectively, as used in Algo. 1; that is $(x_0, x_1) \sim p_0 \times p_1$. Recent works, such as (Pooladian et al., 2023; Liu et al., 2022; Tong et al., 2023), aim to construct more informative joint distributions by sampling $(x_0, x_1)$ from a coupling $\gamma \in \Gamma(p_0, p_1)$:

$$\mathcal{L}_\gamma(\theta) = \mathbb{E}_{t \sim \text{Unif}[0,1], (x_0, x_1) \sim \gamma} \|u(x_t, t; \theta) - \text{TargetVectorField}(x_0, x_1)\|_p^p \quad (2)$$

where $\Gamma(p_0, p_1)$ denotes the set of all couplings (i.e., joint distributions) with marginals $p_0$ and $p_1$:

$$\Gamma := \left\{ \gamma \in \mathcal{P}(\mathcal{X} \times \mathcal{X}) : \int \gamma(x_0, x_1) \, dx_0 = p_1(x_1) \ \ \forall x_1, \int \gamma(x_0, x_1) \, dx_1 = p_0(x_0) \ \ \forall x_0 \right\} \quad (3)$$

As evident from the definition, $\Gamma$ is a vast set. Although training with any valid coupling theoretically yields a correct vector field, the straightness of the resulting trajectories and the efficiency of the training process can vary significantly depending on the choice of coupling. This naturally raises the question: which coupling should we choose? A growing body of work suggests that OT offers a principled way to guide this selection (Pooladian et al., 2023; Kornilov et al., 2024).

### 3.3   SEMI-DISCRETE OPTIMAL TRANSPORT

The Optimal Transport (OT) problem seeks to compute the optimal coupling between two probability distributions given a cost function $c : \mathcal{X} \times \mathcal{X} \to \mathbb{R}$, (see, e.g., Peyré et al. (2019) for a comprehensive overview):

$$\gamma_* := \arg \min_{\gamma \in \Gamma(q_1, q_2)} \left( \int_{\mathcal{X} \times \mathcal{X}} c(y_1, y_2) \, d\gamma(y_1, y_2) \right), \quad (4)$$

where we choose $c(y_1, y_2) := \|y_1 - y_2\|^2$ throughout the paper. [4]

A discrete distribution contrasts with a continuous distribution (such as the normal distribution), in that the associated random variable assumes only a finite (or countable) set of values. For example,

---

[3] We here choose the loss function to be $\text{Loss}\left(\{\hat{v}_t^i, v_t^i\}_{i=1}^B\right) := \frac{1}{B} \sum_i \|\hat{v}_t^i - v_t^i\|_p^p$ for simplicity.

[4] The minimum objective is $W_2^2(q_1, q_2) := \min_{\gamma \in \Gamma(q_1, q_2)} \left( \int_{\mathcal{X} \times \mathcal{X}} \|y_1 - y_2\|^2 d\gamma(y_1, y_2) \right)$.

the empirical data distribution can be expressed as $p_1 = \frac{1}{|I|} \sum_{i \in I} \delta_{x_1^i}$, where $\delta_{x_1^i}$ denotes the Dirac measure centered at data point $x_1^i$. An OT problem between a continuous and a discrete distribution is referred to as Semi-Discrete Optimal Transport (SDOT) (Peyré et al., 2019, Sec. 5), which will serve as our primary technique for aligning noise with data.

The transport map of an SDOT problem can be computed using a $|I|$-dimensional vector $\mathbf{g} = [g_i]_{i \in I}$, referred to as the dual weight, where $|I|$ is the number of points in the discrete distribution $p_1$. Given the dual weight $\mathbf{g}$, we compute $\varphi(\cdot; \mathbf{g}) : \mathcal{X} \to I$, which maps noise to data index:

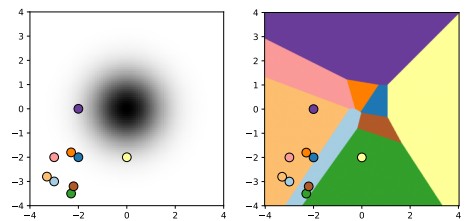

$$\varphi(x_0; \mathbf{g}) := \arg \min_{i \in I} \; c(x_0, x_1^i) - g_i \qquad (5)$$

Given $\varphi(\cdot; \mathbf{g}) : \mathcal{X} \to I$, the SDOT map from the noise distribution to the discrete data points is directly obtained.

(a) Dataset and noise distribution    (b) Laguerre cells

In fact, $\mathcal{X}$ is partitioned into cells by the SDOT map in the sense that cell $\mathsf{L}_i$ contains the points transported to the $i$-th point in the dataset. Each cell is referred to as the Laguerre cell, defined below:

$$\mathsf{L}_i(\mathbf{g}) := \{x \in \mathcal{X} : c(x, y_i) - g_i \leq c(x, y_j) - g_j, \forall j\} \qquad (6)$$

Fig. 1 visualizes the partition in a 2-dim space.

We proceed with the computation of the dual weight $\mathbf{g}$. SDOT, similar to general OT problems in Eq. (4), is a minimization problem. By analyzing its dual problem, the dual weight can be solved by maximizing the following objective function utilizing the Laguerre cells (see e.g., Eq. 5.7 in Peyré et al. (2019)):

Figure 1: Visualization of Laguerre cells in 2-dim. The noise distribution is the normal distribution (dark shadow in the left figure), and the dataset is the points in the lower left corner. The whole space is partitioned into cells using SDOT, and each region is mapped to the data point with the corresponding color by the SDOT map. The integral of the noise density over each cell $\mathsf{L}_i$ equals the discrete probability mass of the associated data point.

$$\mathcal{E}(\mathbf{g}) := \sum_{i \in I} \int_{\mathsf{L}_i(g)} (c(x, y_i) - g_i) \, \mathrm{d}p_0(x) + \langle \mathbf{g}, \mathbf{b} \rangle \qquad (7)$$

whose gradient is given by $\nabla \mathcal{E}(\mathbf{g})_i = - \int_{\mathsf{L}_i(\mathbf{g})} \mathrm{d}p_0 + b_i$, where $b_i$ is the probability for each data point. In our case, $b_i \equiv \frac{1}{|I|}$.

To solve this maximization problem, we employ the Adam optimizer (Kingma & Ba, 2014). While this approach to the SDOT problem is not novel (e.g., Peyré et al. (2019)), we introduce a new and efficient Exponential Moving Average (EMA)-based estimation method for computing the MRE and the $\mathsf{L}_1$ distance (further discussed in Sec. B) to evaluate the quality of the resulting dual weights. This estimation facilitates hyperparameter tuning.

**Computational cost of Algo. 2** : The computation of SDOT dual weight requires $\mathcal{O}(|I|^3)$ iterations to achieve a satisfactory MRE when $\epsilon = 0$ (Liu et al., 2021), and can be further reduced by setting the entropic regularization $\epsilon > 0$, as a positive $\epsilon$ improves the smoothness of the SDOT objective. Specifically, the outer loop (line 4) in Algo. 2 has a per iteration cost of $\mathcal{O}(|I|)$, and the SDOT objective error decreases in the order $\mathcal{O}(1/k)$ (Taşkesen et al., 2023).

### 3.4 THE PROPOSED ALIGNFLOW

OT optimization remains a core challenge in previous OT-based FGMs. In this subsection, we propose using SDOT for Noise-Data Alignment (NDA) in training FGMs.

**Idea behind AlignFlow** The critical insight behind AlignFlow is to use the SDOT to compute a transport map between the noise distribution $p_0$ and the known empirical data distribution $p_1$. This is because 1) SDOT considers all empirical data points during optimization, rather than relying on samples drawn from both the source and target distributions as required by discrete OT; 2) the SDOT

---

**Algorithm 2:** SDOT dual weight optimization

---

1 Input: Source distribution $p_0$, dataset $\{x^i\}_{i \in I}$ and the corresponding probabilities $\mathbf{b} = [b^i]_{i \in I}$, entropic regularization strength $\epsilon$, EMA parameter $\beta$, batch size $B$, cost function $c$
2 Output: Dual weight $\mathbf{g} = [g_i]_{i \in I}$
3 Initialization: $\nabla \mathcal{E}_{\text{ema}} = \mathbf{0}, \mathbf{g} = \mathbf{0}, \mathbf{g}_{\text{ema}} = \mathbf{0}$
4 for $k = 1, 2 \ldots$ do
5      Sample i.i.d. $\{x_0^j\}_{j=1}^B \sim p_0$          ▷ sample noise
6      for $j = 1, \ldots, B$ do
7          if $\epsilon \neq 0$ then
8              $h_j = \text{SoftMax}_{i \in I} \left( -\frac{c(x_0^j, x_1^i) - g_i}{\epsilon} \right)$      ▷ SDOT map with current **g**
9          else
10              $\varphi(x_0^j; \mathbf{g}) = \arg\min_{i \in I} \; c(x_0^j, x_1^i) - g_i$
11              $h_j = \mathbb{1}_{\varphi(x_0^j; \mathbf{g})}$
12      $\nabla \mathcal{E}(\mathbf{g}) = \frac{1}{B} \sum_j h_j - \mathbf{b}$
13      $\nabla \mathcal{E}_{\text{ema}} = \beta \nabla \mathcal{E}_{\text{ema}} + (1 - \beta) \nabla \mathcal{E}(\mathbf{g})$      ▷ Smoothen by EMA
14      Update **g** using $\nabla \mathcal{E}_{\epsilon}(\mathbf{g})$      ▷ optimization
15      $\mathbf{g}_{\text{ema}} = \beta \mathbf{g}_{\text{ema}} + (1 - \beta) \mathbf{g}$
16 Return: dual weight $\mathbf{g}_{\text{ema}}$

---

objective is convex, and the number of iterations required to optimize SDOT is $\mathcal{O}(|I|^3)$ (Liu et al., 2021); and 3) the optimality of the resulting SDOT map can be estimated with sample complexity of $\mathcal{O}(|I|)$. Importantly, the SDOT map between $p_0$ and $p_1$ preserves the desirable properties of OT, such as inducing straight transport trajectories. Therefore, unlike previous OT-based FGMs, AlignFlow solves the SDOT map with a convergence guarantee, leading to an explicit coupling of noise and data.

Once the SDOT map is obtained, it can be integrated into the general FGM framework. This leads to the development of AlignFlow (Algo. 3), which leverages the SDOT map to align noise with data. Notably, Algo. 2 incorporates all data points during optimization and inherently avoids the curse of dimensionality.

---

**Algorithm 3:** AlignFlow: Noise-Data Alignment using SDOT (Training)

---

1 Input: Source distribution $p_0$, dataset $\{x_1^i\}_{i \in I}$, neural network $u(x, t; \theta)$
2 Output: Learned velocity field $u(\cdot, \cdot; \theta)$
3 Hyperparameters: batch size $B$, number of steps $K$
4                      * Stage 1: compute SDOT map *
5 Run Algo. 2 to get dual weight **g**.

6                    * Stage 2: Train flow-based generative model *
7 Let $M = K \cdot B$      ▷ Total number of samples needed
8 Sample i.i.d. $\{x_0^j\}_{j=1}^M \sim p_0$      ▷ sample noise
9 $m_j = \varphi(x_0^j)$ for $j = 1, \ldots, M$      ▷ match noise to data
10 $t^j \sim \mathcal{U}(0, 1)$ for $j = 1, \ldots, M$      ▷ sample time
11 $\{m_j\}_{j=1}^M = \text{Rebalance}\left(\{m_j\}_{j=1}^M\right)$      ▷ Only if needed. Sec. C
     ▷ Training loop
12 for $k = 1 \ldots K$ do
13      for $l = 1 \ldots B$ do
14          $j = (k - 1) \cdot B + l$
15          $x_t^j \leftarrow (1 - t^j) x_0^j + t^j x_1^{m_j}$
16          $v^j = \text{TargetVectorField}(x_0^j, x_1^{m_j})$
17          $\hat{v}^j = u(x_t^j, t^j; \theta)$
18      $L(\theta) = \text{Loss}\left(\{\hat{v}^j, v^j\}_{j=(k-1) \cdot B + 1}^{k \cdot B}\right)$      ▷ objective function
19      Update $\theta$ using $\nabla_\theta L$      ▷ optimization

---

### 3.5 ADDITIONAL TECHNIQUES

**Remark 1** (Noise storage). *In lines 9 and 11 of Algo. 3, a large number of noise samples need to be generated prior to training an FGM. Storing all these samples in memory, or even on disk, quickly becomes impractical.* [5] *To address this, we only store the random seed used to generate each noise sample, so each noise-data pair is represented as a (seed, index) tuple.*

*Such an approach requires a deterministic mapping from the seed to random matrices (supported by JAX) and loading the entire ImageNet latent representations into memory for efficient random access (automatically optimized by the PyTorch dataloader). Using this strategy, storing the (seed, index) pairs for 500 epochs of ImageNet training requires only about 1 GB of disk space.*

**Remark 2** (Data augmentation). *Data augmentation is a critical component for achieving optimal performance in image-related tasks. However, incorporating complex augmentation techniques, such as random cropping or rotation, directly into the SDOT map formulation can be challenging.*

*Fortunately, for most image generation tasks, effective augmentation is often limited to random horizontal flips. This particular case can be handled elegantly without requiring complex modifications to the SDOT framework: simply redefine the dataset as the union of the original images and their horizontally flipped versions.*

**Remark 3** (Class-conditioned generation). *For class-conditioned tasks, as discussed in Sec. 5.2 and 5.3, we denote the class-conditioned data distribution by $p_{1,c}$ for the c-th class. The procedure involves computing the SDOT map between the noise distribution $p_0$ and each class-specific distribution $p_{1,c}$. Once noise samples are drawn, AlignFlow generates corresponding (noise, data) pairs and performs the required Rebalance (Sec. C) operation independently for each class.*

## 4 ADVANTAGES OF ALIGNFLOW

Inspired by the properties of the SDOT map, we introduce the concept of Noise-Data Alignment (NDA):

**Definition 1** (Noise-Data Alignment). *Given a dataset and a noise distribution, noise-data alignment is a deterministic map that maps noise to a data point in the dataset.*

A NDA is a deterministic coupling between noise and data, and we highlight determinism as the key distinction from existing coupling methods, based on the following intuitions: 1) A deterministic coupling ensures consistent noise-data matching regardless of batch size in training FGMs; 2) If $q_1$ in Eq. 4 is continuous, then the OT map from $q_1$ to $q_2$ must be deterministic (see e.g., Peyré et al., 2019, Remark 2.24), i.e., determinism is a necessary condition for optimality.

### 4.1 BYPASS THE CURSE OF DIMENSIONALITY

The sample complexity of OT measures the accuracy of an OT plan estimated by the discrete-OT plan between samples. However, the following theorem shows the number of samples required is exponentially dependent on the dimension:

**Theorem 1** (Sample complexity of OT (Informal version for Thm. 1 in Fournier & Guillin (2015))). *In a $d$-dimensional space, the error in estimating the 2-Wasserstein distance $W_2^2(\tilde{q}, q_n)$ between a distribution $\tilde{q}$ and the empirical distribution $q_n$ with only access to $n$ samples is of order $\sim n^{-1/d}$.*

The squared Wasserstein distance $W_2^2$ corresponds to the minimum value in Eq. (4). Thm. 1 shows that even estimating an OT plan between a continuous distribution and its samples can be challenging, requiring an exponential number of samples. Existing algorithms may suffer from this curse of dimensionality in two cases, potentially both at once: (1) when using samples from the noise distribution to approximate the underlying continuous noise distribution; and (2) when using the dataset or a batch from the dataset to approximate the unknown real data distribution. Many works (implicitly) assume that the empirical distribution approximates the continuous one, e.g., Assumption A2 and A3 in Pooladian et al. (2023), Thm.1 in Kornilov et al. (2024).

---

[5] Based on our experiments, storing 10 epochs worth of noise samples for ImageNet training in latent space (see Sec. 5.3 and 5.2) would require terabytes of disk space. In addition to storage demands, I/O throughput would pose a significant bottleneck.

As a NDA approach, AlignFlow circumvents the curse of dimensionality by computing an SDOT map between $p_0$ and $p_1$. Since $p_1$ is fully specified by the dataset, the SDOT map can, in principle, be computed with zero estimation error, thereby avoiding the curse of dimensionality.

## 4.2 DETERMINISTIC ALIGNMENT

The SDOT map is theoretically fully deterministic: a given sample from the noise distribution is consistently mapped to a fixed data point.

Intuitively, this determinism provides a significant advantage in convergence speed: to determine the target vector field $v_t$ at some $t$ and $x_t$, the standard approach using the random coupling (as in Algo. 1) requires iterating over the entire dataset:

$$u(x_t, t; \theta) = \mathbb{E}_{x_1 \sim p_1} \text{TargetVectorField}(x_0, x_1) p_0(x_0), \quad x_0 = x_1 - (x_1 - x_t)/t \quad (8)$$

However, the fixed coupling in AlignFlow bypasses this estimation process, directly providing the target vector field for the NN to learn as:

$$u(x_t, t; \theta) = \text{TargetVectorField}(x_0, x_1) p_0(x_0), \quad x_0 = x_1 - (x_1 - x_t)/t, \ x_1 = \varphi(x_0) \quad (9)$$

This crucial difference demonstrates that fixed coupling significantly simplifies the estimation of the target vector field, thereby leading to the accelerated convergence observed empirically with AlignFlow.

## 4.3 LOW COMPUTATIONAL COST

AlignFlow directly computes the SDOT map in Stage 1. In contrast to prior methods that estimate the OT plan indirectly through batch-based approaches (e.g., Reflow operation in Liu et al. (2022), ICNN in Kornilov et al. (2024), and Minibatch OT in Tong et al. (2023)). AlignFlow's direct computation of SDOT is both more accurate and efficient, and Stage 1 adds negligible overhead empirically, accounting for less than 1% additional training time. Further implementation details are provided in Sec. B.

Upon completion of Stage 1, the SDOT map is fully computed. Consequently, the only additional overhead in Stage 2 arises from generating the training noise-data pairs (Lines 9 and 11 in Algo. 3). This process is highly efficient, runs quickly on modern GPUs, and incurs an almost negligible cost, typically less than 0.1% of the total training time.

## 5 EXPERIMENTS

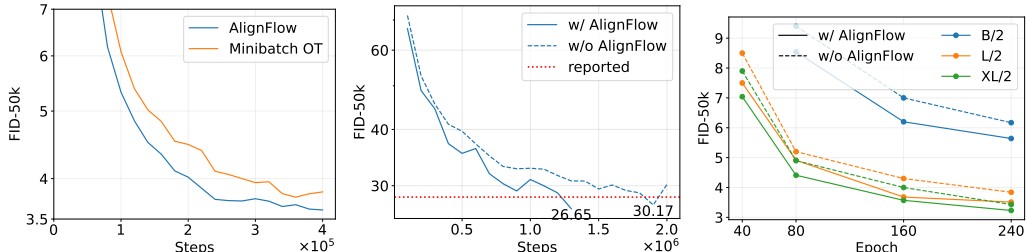

(a) CIFAR-10 results using U-Net, generated by the adaptive ODE integrator DOPRI5 (see Sec. 5.1).

(b) DiT-B/2 on ImageNet256 with the shortcut model, generated using 4-step forward Euler.

(c) SiT on ImageNet256 with MeanFlow, generated using 1-step forward Euler.

Figure 2: Training curves for AlignFlow on different tasks. Each figure illustrates the FID-50k score against the number of training steps. The results demonstrate that AlignFlow provides a consistent and simultaneous improvement over all baseline algorithms shown, enhancing both final performance and training convergence speed.

## 5.1 UNCONDITIONAL IMAGE GENERATION ON CIFAR-10 WITH U-NET

Following (Tong et al., 2023, Sec. 5.3), we train an unconditional U-Net (Ronneberger et al., 2015) on the CIFAR-10 dataset. In this setup, the FGMs are trained directly in the pixel space. The com-

parative training curves are shown in Fig. 2(a), and the FID-50k scores for various ODE integrators are detailed in Table 2. Compared to the coupling estimated via the standard Minibatch OT (Tong et al., 2023), AlignFlow demonstrates faster convergence and achieves better FID scores across all tested ODE integrators. All experiments were performed using the official code released by Tong et al. (2023).

|  | Euler (100 steps) | Euler (1000 steps) | DOPRI5 |
|---|---|---|---|
| Minibatch OT (Tong et al., 2023) | 4.80 | 3.92 | 3.82 |
| AlignFlow (ours) | 4.72 | 3.79 | **3.71** |

Table 2: Comparison of FID-50k scores between Minibatch OT and AlignFlow for U-Net trained on CIFAR-10, evaluated across different ODE integrators. Results are averaged over 5 independent runs. AlignFlow consistently achieves better performance than Minibatch OT under all tested ODE integrators.

## 5.2 IMAGENET256 GENERATION USING DIT WITH SHORTCUT MODEL

AlignFlow can be seamlessly integrated with modern SOTA network architectures and scales effectively to large-scale datasets. We train AlignFlow in conjunction with Diffusion Transformers (DiT, Peebles & Xie (2023)) on the class-conditioned ImageNet dataset at $256 \times 256$ resolution (ImageNet256). The FGMs operate in the latent space of size $28 \times 28 \times 4$, obtained from a pretrained VAE. All model hyperparameters are adopted directly from (Frans et al., 2025, Tables 1 and 3) without any modification or tuning. A comparison of training curves for the shortcut model with and without AlignFlow is shown in Fig. 2(b), while improvements across additional models using AlignFlow are reported in Table 3.

| Algorithm | AlignFlow? | NFE=4 | Difference | NFE=1 | Difference |
|---|---|---|---|---|---|
| Flow Matching | ✓ | 93.16 | ↓ 32.46 | 276.18 | ↓ 28.86 |
|  | ✗ | 125.62 |  | 305.04 |  |
| Consistency Training | ✓ | 103.14 | ↓ 8.70 | 64.33 | ↓ 12.04 |
|  | ✗ | 111.84 |  | 76.37 |  |
| Live Reflow (Frans et al., 2025) | ✓ | 60.23 | ↓ 34.52 | 47.06 | ↓ 12.81 |
|  | ✗ | 94.75 |  | 59.87 |  |
| Shortcut Models (Frans et al., 2025) | ✓ | **30.31** | ↓ 2.80 | **43.92** | ↓ 2.73 |
|  | ✗ | 33.11 |  | 46.65 |  |

Table 3: Evaluation of AlignFlow on DiT-B/2 for ImageNet256 using FID-50k demonstrates consistent performance improvements across all tested NFE configurations.

## 5.3 IMAGENET256 GENERATION USING SIT WITH MEANFLOW

AlignFlow further enhances the one-step generative model MeanFlow (Geng et al., 2025), which is based on Scalable Interpolant Transformers (SiT, Ma et al. (2024)) and trained on class-conditioned ImageNet256. The FGMs operate in the latent space of size $28 \times 28 \times 4$, obtained from a pretrained VAE. For our experiments, we use a non-official PyTorch implementation (Zhu, 2025), which reliably reproduces the reported results on GPU. All hyperparameters are kept identical to the official configuration (Geng et al., 2025, Section A), without any additional tuning. The training curve is shown in Fig. 2(c), and the corresponding FID scores are reported in Table 4. AlignFlow consistently improves both performance and convergence speed across all cases, demonstrating its scalability to large models. Example generated images are provided in Fig. 4 in the appendix.

| Backbone | # params | w/ AlignFlow | w/o AlignFlow | Difference |
|---|---|---|---|---|
| SiT-B/4 | 131M | **13.75** | 15.53 | ↓ 1.78 |
| SiT-B/2 | 131M | **5.60** | 6.17 | ↓ 0.57 |
| SiT-L/2 | 459M | **3.51** | 3.84 | ↓ 0.33 |
| SiT-XL/2 | 676M | **3.23** | 3.43 | ↓ 0.20 |

Table 4: We evaluate AlignFlow on ImageNet256 using MeanFlow (NFE=1), showing consistent performance improvements across all model sizes.

ACKNOWLEDGMENT

LK thanks Amazon for hosting his internship. LK and MT are grateful for partial supports by NSF Grants DMS-1847802, DMS-2513699, DOE Grants NA0004261, SC0026274, and Richard Duke Fellowship. We also thank Macro Cuturi for valuable comments during the review process, which significantly improved the quality of this work.

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

# A FUTURE WORK

Many modern tasks involve datasets with complex label structures, such as text-to-image generation where each data point is a tuple consisting of an image and text. Since text prompts are rarely identical across a dataset, standard label-based techniques are insufficient.

Here we can provide an idea of how AlignFlow may be used to address this task: suppose the data is given by tuples $(x, y)$, where $x$ is the image (let's take the image modality as an example) and $y$ is the text, and the task is to train AlignFlow that generates new $\tilde{x}$ given some new $\tilde{y}$. We can cluster the text $y$ (e.g., via an LLM or extract the text embedding of each text $y$ and apply a traditional clustering method) and assign a label $z = \text{clustering}(y)$ to each $y$, making the input data $(x, y, z)$. For example, $y_1 =$"a dog is swimming" and $y_2 =$"a dog is running" could be clustered into the same cluster. Then, for each cluster, we compute the SDOT map $\varphi|_z$ to map noise to images $x$. Training the flow-based generative model with $v(x|y)$ can be guided by the corresponding SDOT map $\varphi|_z$, where $z$ is the class label for $y$.

# B EFFICIENT SDOT ALGORITHM FOR LARGE-SCALE DATASETS

## B.1 PERFORMANCE METRIC FOR ALGO. 2

We revisit the definitions of Maximum Relative Error (MRE) and $L_1$ distance as proposed in (Liu et al., 2021), which serve as metrics for evaluating the quality of the computed dual weights:

$$\text{MRE} = \max_{i \in I} \frac{|p_i - b_i|}{b_i}, \quad L_1(\mathbf{g}) = \sum_{i \in I} |p_i - b_i|, \quad p_i := \int \mathbb{L}_i(\mathbf{g}) \, dp_0 \tag{10}$$

In Algo. 2, they can be estimated efficiently by $\widetilde{\text{MRE}} = \|\nabla \mathcal{E}_{\text{ema}}\|_\infty$ and $\widetilde{L_1} = \|\nabla \mathcal{E}_{\text{ema}}\|_1$.

Mathematically, MRE and $L_1$ serve as estimators for the $\ell_\infty$-norm and $\ell_1$-norm of the gradient of the objective, specifically $\|\nabla \mathcal{E}\|_\infty$ and $\|\nabla \mathcal{E}\|_1$, respectively. The critical importance of MRE lies in its ability to quantify the imbalance of the transported probabilities. Importantly, an outcome of $\text{MRE} < 1$ guarantees that each target data point receives non-zero probability transported from the source distribution. $\text{MRE} = 0$ signifies that the SDOT solution is perfectly optimized: the objective function in Eq. (4) is minimized, and, crucially, the measure preservation constraint is satisfied uniformly, ensuring each target receives an equal probability mass. (See also Sec. C for details on rebalancing.)

## B.2 ENTROPIC REGULARIZATION FOR SDOT

Discrete OT problems are intrinsically non-smooth. A possible approach to mitigate this is the introduction of regularization terms to induce a smoother objective landscape. For SDOT problems, similar techniques can also be applied. Instead of solving the problem Eq. (4), SDOT with entropic regularization is solving (Altschuler et al., 2022)

$$\min_{\gamma \in \Gamma(p_0, p_1)} \int_{\mathcal{X} \times \mathcal{X}} c(x_0, x_1) d\gamma(x_0, x_1) + \epsilon \, \text{KL}(\gamma \| p_0 \otimes p_1) \tag{11}$$

where $\epsilon > 0$ is the regularization strength and KL denotes the Kullback-Leibler divergence. While the inclusion of this auxiliary term introduces a controlled bias into the solution, it significantly enhances the smoothness and tractability of the optimization landscape.

## B.3 HYPERPARAMETERS TUNING FOR SDOT OPTIMIZATION

Algo. 2 involves the tuning of three primary hyperparameters: the entropic regularization strength $\epsilon$, the EMA parameter $\beta$, and the Adam optimizer's learning rate $lr$. Throughout the maximization process in Algo. 2, the $L_1$ metric is expected to exhibit a continuous decay when the hyperparameters are set appropriately.

- Entropic regularization strength $\epsilon$ dictates the fundamental trade-off between bias and optimization difficulty. A larger $\epsilon$ induces a greater bias in the resulting transport plan, while

| # step | learning rate | batch size | EMA parameter $\beta$ | entropic reg $\epsilon$ | MRE | $L_1$ |
|---|---|---|---|---|---|---|
| 0-1000 | 10 | 1024 | 0.99 | 1 | 3.4 | 0.29 |
| 1000-6000 | 0.1 | 4096 | 0.999 | 1 | 0.27 | 0.045 |
| 6000-11000 | 0.1 | 16384 | 0.999 | 0.01 | 0.11 | 0.019 |

Table 5: SDOT hyperparameters for CIFAR-10 (unconditional) generation. Computation takes 8 minutes and 30 seconds on an NVIDIA L40S GPU.

| # step | learning rate | batch size | EMA parameter $\beta$ | entropic reg $\epsilon$ | MRE | $L_1$ |
|---|---|---|---|---|---|---|
| 3000 | 10 | 4096 | 0.99 | 0.01 | $\sim 0.08$ | $\sim 0.016$ |

Table 6: SDOT hyperparameters for ImageNet256 (class-conditioned, latent space) generation. Computation takes less than 10 seconds per class on an NVIDIA L40S GPU.

      setting $\epsilon$ to zero or a very small value significantly increases the non-smoothness and complexity of the optimization landscape.

- The parameter $\epsilon$ should remain fixed throughout the entire optimization procedure. When optimizing the problem, consider increasing the batch size and/or decreasing the learning rate when $L_1$ plateaus.

- The optimization stops when MRE satisfies the required performance threshold. For robust performance in downstream tasks, such as FGMs, we recommend maintaining MRE below 0.2.

- Contrary to common practices in standard deep learning optimization, where learning rates around $0.001$ are preferred, the computation of the SDOT map (Algo. 2) often requires a relatively high initial learning rate. A value of $lr = 10$ serves as a recommended starting point for tuning.

- For the management of large datasets, it is advisable to increase the batch size and/or adjust the EMA parameter $\beta$ closer to 1 (e.g., from 0.99 to 0.999) to ensure a smoother and more stable gradient update over a larger number of iterations.

### B.4 HYPERPARAMETERS AND COMPUTATIONAL COST

On CIFAR-10, we use the training set for training (50000 images with shape $32 \times 32 \times 3$).

- Normalize the whole dataset with mean $= (0.5, 0.5, 0.5)$ and std $= (0.5, 0.5, 0.5)$.
- Concatenate the dataset with the augmented (horizontally flipped) dataset.
- Compute the SDOT map with Algo. 2 and hyperparameters in Table 5

On ImageNet, we use the full training set consisting of 1,281,167 images across 1,000 classes. In both the shortcut model w/ AlignFlow (Sec. 5.2) and MeanFlow w/ AlignFlow (Sec. 5.3), the SDOT map is computed in the latent space of size $28 \times 28 \times 4$. Each image is augmented via horizontal flipping, resulting in approximately 2,600 images per class after augmentation. In our experiments, the SDOT map computation on the ImageNet dataset is cheaper than on the CIFAR-10 dataset. Evaluating the SDOT map $\varphi$ in Eq. (5) (Line 9 in Algo. 3) requires computing the minimum across the entire dataset, incurring a computational cost of $\mathcal{O}(|I|)$ with a small constant factor. While this may initially appear computationally expensive, modern machine learning models are typically over-parametrized, meaning the number of parameters vastly exceeds the number of data points. Consequently, the computational cost is dominated by the forward pass and backward propagation rather than the SDOT map evaluation. This analysis explains why the additional overhead observed in our experiments is negligible, regardless of dataset size.

## C REBALANCE: HANDLING NON-PERFECT SDOT MAP

Algo. 3 deviates from the standard practice of sampling i.i.d. data pairs. Instead, it only samples noise $x_0 \sim p_0$ and subsequently feeds the NN with data generated by the SDOT map (Line 9).

This specific sampling scheme introduces a potential issue: if the SDOT map is not computed perfectly, the data $\varphi(x_0)$ fed to the NN will be biased. Specifically, when the gradient of the energy $\nabla\mathcal{E}$ is non-zero, the set of mapped samples, represented by the set of indices $\{\varphi(x_0) \mid x_0 \overset{\text{i.i.d.}}{\sim} p_0\}$ becomes biased from the empirical data distribution $p_1$. (See also MRE defined in Eq. (10) in Sec. B). This bias means the model will be trained with an uneven exposure to the dataset, with some data points being seen more frequently than others.

To mitigate the effects of imperfect SDOT convergence, a particular concern for large, non-conditional datasets where running Algo. 2 to MRE $\to 0$ is computationally prohibitive, we introduce the rebalance operation in Line 11, defined as:

$$
\text{rebalance}\left(\{m_j\}_{j=1}^M\right) := \arg\max_{\{\tilde{m}_j\}} \left\{ \sum_j \mathbb{1}(\tilde{m}_j = m_j) : \left| \max_{i\in\mathcal{I}} \sum_{j=1}^M \mathbb{1}_i(\tilde{m}_j) - \min_{i\in\mathcal{I}} \sum_{j=1}^M \mathbb{1}_i(\tilde{m}_j) \right| \le 1 \right\}
\tag{12}
$$

Intuitively, the rebalance operation determines the minimal perturbation to the set of target indices $\{m_j\}$ such that the empirical frequency of each data point in the modified set $\{\tilde{m}_j\}$ is nearly uniform. This operation forces the model to be exposed to an effectively unbiased dataset during the FGM training phase, even if the SDOT map is not fully converged. This comes at the cost of introducing a minimal amount of controlled randomness in the noise-data coupling.

In our experiments, particularly on the CIFAR-10 dataset (Sec. 5.1), the rebalance operation did not yield a noticeable difference in performance. This is likely because Algo. 2 achieved a sufficiently high degree of convergence (evidenced by the fact that over $85\%$ of the data assignments remained unchanged after the rebalance operation). Nonetheless, the rebalance mechanism is applied consistently across all our experiments to ensure that the data fed into the FGMs is unskewed.

## D MORE DETAILS FOR FLOW-BASED GENERATIVE MODELS

### D.1 MORE EXAMPLES FOR FGM FRAMEWORK IN ALGO. 1

**Shortcut model** In this model, an auxiliary input $d$ is introduced to the NN, i.e., $u = u(x, t, d; \theta)$ and $d^j$ is i.i.d. sampled from $\mathcal{D}(\cdot|t^j)$. Given hyperparameter $\kappa$, choose $\text{TargetVectorField}(x_0^j, x_1^j) = x_1^j - x_0^j$ for $j = 1, ..., \kappa$, and $\text{TargetVectorField}(x_0^j, x_1^j) = \text{StopGrad}(s_t^j + s_{t+d}^j)$ for $j = \kappa + 1, ..., B$, where $s_t^j := u(x_t^j, t^j, d^j)$, $x_{t+d}^j := x_t^j + s_t d^j$, $s_{t+d}^j := u(x_{t^j+d^j}^j, t^j + d^j, d^j)$. Together with $\text{Loss}\left(\{\hat{v}^j, v^j\}_{j=1}^B\right) := \frac{1}{B}\sum_j \left\|\hat{v}^j - v^j\right\|_2^2$, Algo. 1 recovers the shortcut model in Frans et al. (2025).

**MeanFlow** In this model, an extra input $r$ is introduced to the NN, i.e., $u = u(x, t, r; \theta)$. By choosing $\text{TargetVectorField}(x_0, x_1) = \text{StopGrad}(v_t - (t - r)v_t\partial_x u + \partial_t u)$ and $\text{Loss}\left(\{\hat{v}^i, v^i\}_{i=1}^B\right) := \frac{1}{B}\sum_i \left\|\hat{v}^i - v^i\right\|_p^p$, Algo. 1 recovers the MeanFlow in Geng et al. (2025).

### D.2 SAMPLING / INFERENCE PROCEDURE OF FGMs

To better elucidate the advantages conferred by the specially-designed target vector field, we review the inference procedure for FGM, as outlined in Algo. 4. The inference process entails integrating the ODE $\partial_t x_t = u(x_t, t; \theta)$ over the time interval $t \in [0, 1]$, where the initial condition $x_0$ is sampled from the noise distribution $p_0$, and the final state $x_1$ constitutes a new data sample. While various ODE integrators may be employed, the computational difficulty of this integration is fundamentally governed by the complexity of the learned vector field $u(x, t; \theta)$. Intuitively, consider two hypothetical trajectories: an almost straight and a highly complex, curvilinear path. The almost straight path permits straightforward computation, possibly achievable in a single step via the forward Euler method, $x_1 = x_0 + u(x_0, 0; \theta)$. Conversely, the complex path necessitates a high-order or small-step ODE solver. This intuition highlights a key insight: the **straightness of the integrated trajectory** is paramount. A straighter path implies easier numerical integration, resulting in a lower NFEs of the NN (Liu et al., 2022). Algorithms trained with a carefully-designed target vector field benefit directly from this enhanced straightness. For instance, MeanFlow (Geng et al., 2025) is capable

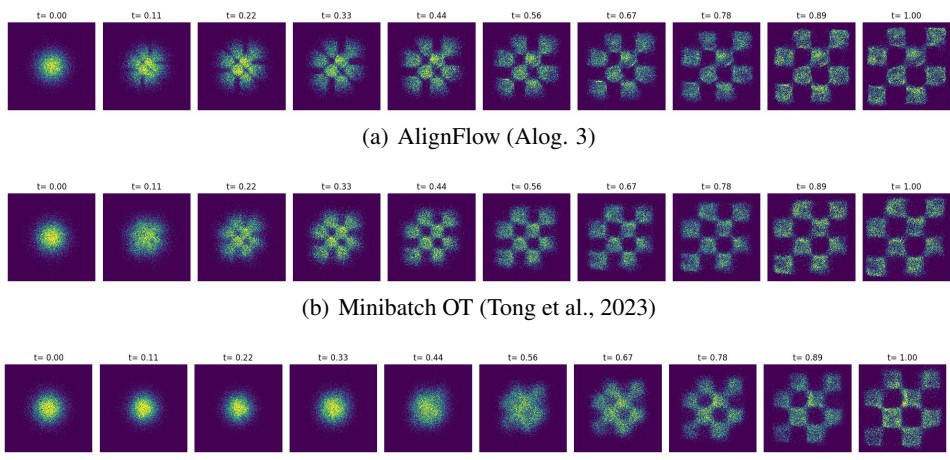

(a) AlignFlow (Alog. 3)

(b) Minibatch OT (Tong et al., 2023)

(c) Vanilla Flow Matching (Lipman et al., 2022)

Figure 3: Comparison of generative trajectories in FGMs among various methods. AlignFlow has a straighter trajectory compared to vanilla Flow Matching and has a clearer boundary compared to Minibatch OT (e.g., at $t = 0.22$).

of generating high-quality samples with an NFEs as low as 1, whereas the vanilla Flow Matching (Lipman et al., 2022) typically requires over 100 NFEs to achieve comparable results.

---

**Algorithm 4:** FGM / AlignFlow (Sampling)

---

1 Input: Noise distribution $p_0$, neural network $u = u(x, t; \theta)$, ODEIntegrator
2 Output: A new sample from the data distribution
3 Sample $\{x(0)\} \sim p_0$         ▷ sample noise
4 $v(t) := u(x(t), t; \theta)$
5 $x(1) = \text{ODEIntegrator}(v(t))$
6 Return: a new sample $x(1)$

---

# E SYNTHETIC EXPERIMENT: CHECKERBOARD

In this section, we analyze and visualize the characteristics of different learned trajectories by training the FGMs algorithm on a synthetic two-dimensional data distribution. Following the precedent set by (Lipman et al., 2022, Fig. 4), the target data distribution is defined as the checkerboard distribution in $[-2, 2] \times [-2, 2]$. The noise distribution $p_0$ is the standard normal distribution.

Crucially, our experimental setting differs from prior work (Lipman et al., 2022; Tong et al., 2023), which assumed access to an infinite data stream during training.[6] In our experiments, we sample data from the checkerboard distribution, fix the samples, and use them as the training set. While this fixed-data approach may result in a learned distribution with less spatial smoothness compared to the infinite-data setting, it provides a more faithful simulation of real-world machine learning tasks where data availability is inherently limited.

Fig. 3 illustrates the density evolution as time $t$ progresses from 0 to 1, charting the transformation from the Gaussian distribution $p_0$ to the checkerboard distribution. The visualizations indicate that AlignFlow generates a significantly straighter transport trajectory when compared to trajectories learned via Minibatch OT and Vanilla Flow Matching, as depicted in the corresponding panels of Fig. 3. All comparative experiments utilize identical hyperparameters to ensure a fair comparison of the learned vector fields.

---

[6]In the checkerboard experiments of Lipman et al. (2022); Tong et al. (2023), fresh training data is sampled for every minibatch.

## F   CAPABILITY OF GENERALIZATION

A natural question arises from the observation that the SDOT map already constitutes a mapping from the source space to the target space ($\mathcal{X} \to \mathcal{X}$). Since the integrated vector field of an FGM also provides an $\mathcal{X} \to \mathcal{X}$ transformation, why is the subsequent FGM training in Stage 2 still necessary? The fundamental limitation of the SDOT map is its inability to generalize: it is constrained to only map noise samples to the fixed points in the dataset. Consequently, it cannot generate new data instances. Therefore, a powerful NN needs to be trained in Stage 2 of Algo. 3 atop the SDOT alignment to acquire the necessary generalization capability. The observed generalization performance of standard FGMs suggests that AlignFlow's generalization performance can be achieved from the inductive bias and regularization inherent in the NN used to learn the vector field.

# G    IMAGE SAMPLES GENERATED BY ALIGNFLOW

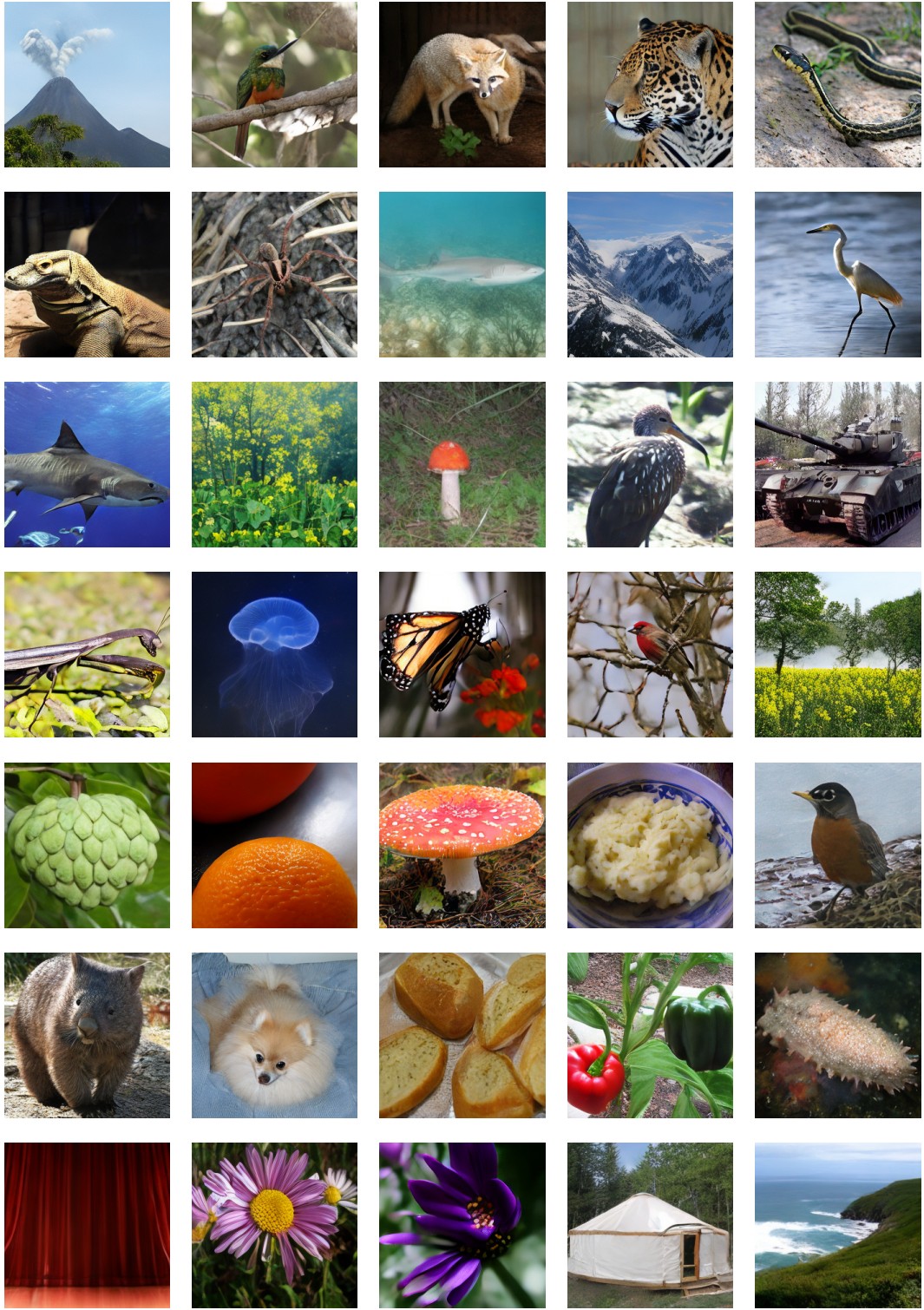

Figure 4: Images generated by MeanFlow+AlignFlow trained on ImageNet256 (FID=3.23, NFE=1).

