# OpenReview forum: "AlignFlow: Improving Flow-based Generative Models with Semi-Discrete Optimal Transport"
_ICLR.cc/2026/Conference — ICLR 2026 Poster_

### Official Review · Reviewer_mCfS · 2025-10-15

**Soundness:** 3
**Presentation:** 3
**Contribution:** 3
**Rating:** 6
**Confidence:** 3

**Summary:**

AlignFlow introduces semi-discrete optimal transport (SDOT) as a deterministic way to align each noise-data pairs to ensure more straight flow paths in flow-based generative models (FGMs) training, reducing Number of Function Evaluations (NFE).

A dual-weight vector defines the transport plan between noise and data, corresponding to Laguerre cells that partition the noise space. Computes dual-weight vector by using Adam optimizer in maximizing an objective from the dual problem.

Training proceeds in two stages: (i) compute the SDOT map once (extra cost < 1 %), and (ii) train any FGM while re-using these fixed (noise, data) pairs .

Deterministic alignment yields straighter generative paths, fewer NFE, and bypasses the sample-complexity curse.

Plug-and-play integration with Flow-Matching, Shortcut Models, MeanFlow, etc. Improves FID across CIFAR-10 and ImageNet-256 at low NFE.

**Strengths:**

The paper is written clearly. It explains the problem settings, methodology, and contributions quite well.

Uses SDOT to build a deterministic coupling is a novel yet simple idea that avoids high-dimensional OT pitfalls. Similar ideas are rarely explored and has only one contemporary work published in 2025.

Demonstrated improvement across model sizes and datasets to imply the generality of the algorithm. Wall-clock times for experiment runs on CIFAR and ImageNet are reported to support claims for SDOT cost.

**Weaknesses:**

1. Although recommendations for setting SDOT hyper-parameters ($\epsilon$, lr, EMA β) is discussed qualitatively to help practitioners use the algorithm, sensitivity to hyperparameters is not discussed in the experiments. If the algorithm is to be further scaled, existing practices for choosing hyper-parameters could potentially fail.
2. Experiments are done on a rather limited data set size; scalability to larger datasets or infinitely sampled settings (e.g., synthetic data augmentation) is unclear. Paper mentioned that processing time is 8min 30s for CIFAR10, and that the algorithm scales quadratically w.r.t. number of targets. Extrapolating to industrial-sized datasets with typically millions of samples, the pre-processing could be astoundingly time-consuming.
3. The difference of FID-50k with and without AlignFlow for larger models and better FID is quite small according to Table 3. Even though AlignFlow improves FID across sizes, the improvement is not very significant. Perhaps the authors should consider using a new evaluation task or metric to highlight improvement.

**Questions:**

1. The paper states that "AlignFlow particularly suitable for large models that require small-batch training". This seems to weaken the contribution of this paper, as methods such as [1] works on small-batch training and [2] works on larger models already.

2. Could authors explain the reasons behind excluding discussions and experiments of AlignFlow on continuous normalizing flows (CNFs)? Demonstrating benefits there would strengthen the claim of generality.

3. Could the SDOT map be updated online to accommodate new data without recomputing from scratch?

##### Reference

[1] Alexander Tong, Kilian Fatras, Nikolay Malkin, Guillaume Huguet, Yanlei Zhang, Jarrid RectorBrooks, Guy Wolf, and Yoshua Bengio. Improving and generalizing flow-based generative models with minibatch optimal transport.

[2] Zhengyang Geng, Mingyang Deng, Xingjian Bai, J Zico Kolter, and Kaiming He. Mean flows for one-step generative modeling.

---

> ### Author Response · Authors · 2025-11-21
> **Response Part 1/2**
>
> We appreciate the reviewer for confirming the novelty and the good performance of the proposed algorithm.
>
> Weaknesses:
>
> > 1. Sensitivity to hyperparameters for computing the SDOT map.
>
> We acknowledge the reviewer's concern regarding hyperparameter sensitivity. Here, we respectfully emphasize a fundamental distinction between our approach and standard deep learning training: the SDOT optimization problem is convex. Unlike non-convex neural network training, where hyperparameters dictate which local minimum is reached, thereby affecting final performance. The convexity of the entropic-regularized semi-discrete OT problem guarantees a unique global optimum and the convergence of SGD. Consequently, hyperparameters such as learning rate, $\beta$, and batch size influence only the convergence speed (efficiency), not the quality of the final map. In practice, we followed the routine in Sec. A.3 solely to identify the most efficient configuration.
>
>
> > 2. Computation cost for SDOT map computation in industrial-sized datasets.
>
> As the reviewer points out, scalability is important for computing the SDOT map. The reviewer is correct that it is difficult to compute the map with millions of data points directly. However, this is mitigated by class conditioning: we have shown that unconditional CIFAR can be handled, and for huge datasets, as long as each class has fewer than 50,000 images (the number of images in CIFAR), the SDOT map can be computed efficiently. We believe this is sufficient for many modern tasks, since large, industrial-level datasets are often class-conditioned (e.g., ImageNet has about 1,000 images per class).
>
> > 2. infinitely sampled settings (e.g., synthetic data augmentation)
>
> For synthetic data augmentation methods that generate an infinite number of samples for each original sample, one possible way to handle this is as follows. First, we compute the SDOT map using only the original (non-augmented) samples in the dataset. During FGM training, we then sample noise and use the SDOT map to find the corresponding image. This gives us the original image; we then pair its augmented version with the sampled noise to train the FGM. This approach relies on the assumption that an original image is close to its augmented versions in feature space, which is generally considered reasonable.
>
>
> > 3. Small improvement from AlignFlow for larger models and better FID
>
> We thank the reviewer for this observation. We agree that as baseline models approach state-of-the-art performance, the margin for further enhancement naturally narrows. However, while the absolute reduction in FID may appear marginal, the relative improvement remains significant. For instance, the 0.2 reduction in FID reported for our best model (SiT-XL/2 in Table 4) corresponds to a substantial 6% relative improvement given the strong baseline.

---

> ### Author Response · Authors · 2025-11-21
> **Response Part 2/2**
>
> Questions:
> > 1. The paper states that "AlignFlow particularly suitable for large models that require small-batch training". This seems to weaken the contribution of this paper, as methods such as [1] works on small-batch training and [2] works on larger models already.
>
> We thank the reviewer for pointing this out. We have revised to "This batch-invariance ensures stable convergence even when batch sizes are severely constrained, addressing a critical bottleneck for training large-scale models where memory limitations enforce small-batch regimes." Our intention was not to limit the contribution of AlignFlow to small-batch settings, but rather to highlight that its mathematical properties (specifically, the deterministic nature of the SDOT map) make it robust to random sampling of noise that typically degrades performance in such settings. We have revised the text to clarify that the batch-invariance of the SDOT map is a fundamental theoretical contribution that results in superior stability. This stability is universally beneficial, but becomes the critical differentiating factor when hardware constraints force small-batch training.
>
> > 2. The reason for excluding continuous normalizing flows (CNFs)
>
> The CNFs we meant are traditional CNFs that rely on Likelihood Maximization. The training objectives of the traditional CNFs are completely different from FGM objectives, and thus, AlignFlow is mathematically not applicable to the traditional CNFs.
>
>
> > 3. Could the SDOT map be updated online to accommodate new data without recomputing from scratch?
>
> We admit our current framework cannot handle the online training, since computing the SDOT plan requires the complete dataset. We appreciate the reviewer for pointing this out and this is very interesting future work. Our current idea for achieving this is as follows. When a new data point arrives, we update the SDOT plan. The new dual weight of the new data point is initialized as the dual weight of the nearest neighbor of the new data point in the dataset. Based on the intuition that the dual weight for SDOT plan will only slightly change with only one new data point, the update for SDOT plan could possibly be cheap.

---

> > ### Comment · Reviewer_mCfS · 2025-11-27
> >
> > Thanks for the response. I will keep my rating as is.

---

### Official Review · Reviewer_GdTu · 2025-10-30

**Soundness:** 3
**Presentation:** 3
**Contribution:** 3
**Rating:** 6
**Confidence:** 3

**Summary:**

The paper proposes AlignFlow, a two-stage recipe for FGM training: (1) precompute a semi-discrete OT (SDOT) map from continuous noise to the empirical data distribution by partitioning the noise space into Laguerre cells; (2) train any flow-matching-style model while pairing each sampled noise with its SDOT-assigned datum. The method is “plug-and-play,” deterministic (fixed coupling), and reported to (a) speed convergence and (b) improve FID across several FGM families (Flow Matching, Consistency/Shortcut, Live Reflow, MeanFlow) on CIFAR-10 and ImageNet256 (latent). The paper also argues that SDOT avoids the OT sample-complexity “curse of dimensionality” by optimizing against p_1 rather than the unknown population \tilde p_1.
	​

**Strengths:**

(1) Simple, general mechanism: drop-in coupling that works with multiple FGM targets (FM, Shortcut, MeanFlow, etc.). Algorithm 3 is clear.
(2) Practical engineering: seed-based noise storage; class-wise SDOT; lightweight Stage-1 overhead; handy MRE/L1 diagnostics and tuning guidance.
(3) Empirical gains: better FID at fixed NFE across multiple backbones; improvements persist from U-Net/CIFAR10 to DiT/SiT on ImageNet256. Tables 2–4 & Fig. 2.

**Weaknesses:**

(1) The paper acknowledges no assumption about SDOT ≈ population OT, but several claims and intuitions (e.g., straighter, “more optimal” paths) implicitly rely on population properties.
(2) Some parts of the presentation are unclear, e.g. the algorithm description.

**Questions:**

(1) Does the method resample noise every epoch?

---

> ### Author Response · Authors · 2025-11-21
>
> Thank you for confirming the novelty, empirical improvement and theoretical benefit of the paper!
>
> Weaknesses
>
> > (1) The paper acknowledges no assumption about SDOT ≈ population OT, but several claims and intuitions (e.g., straighter, “more optimal” paths) implicitly rely on population properties.
>
> Sorry for the confusion. By 'straighter' and 'more optimal', we meant that AlignFlow guided by SDOT produces a straighter and more optimal path than the paths produced by sample-based OT algorithms. For example, the OT plan in sample-based OT depends highly on the minibatch and varies in each iteration during training, meaning the OT plan in each minibatch is biased. This leads to a non-straight path for training of FGMs in expectation.
>
> > (2) Some parts of the presentation are unclear, e.g. the algorithm description.
>
>
> Thanks for your suggestion to improve the quality of this paper! We have proofread this paper and fixed the parts that were not clear. We have added a description for Algorithm 2 at the end of Sec. 3.3 in page 6.
>
>
> Questions:
> > (1) Does the method resample noise every epoch?
>
> Yes. We pre-sample a large set of noise at the beginning of training and ensure that samples are never reused. Consequently, the noise samples are unique to each epoch and are utilized only once throughout the entire training process.

---

> > ### Comment · Reviewer_GdTu · 2025-11-23
> >
> > I appreciate the authors’ detailed response, which addressed my remaining concerns. I will keep my rating unchanged.

---

### Official Review · Reviewer_urUF · 2025-10-31

**Soundness:** 3
**Presentation:** 3
**Contribution:** 3
**Rating:** 6
**Confidence:** 4

**Summary:**

The paper proposes AlignFlow, a method that calculates semi-discrete optimal transport (SDOT) between the source and the data distributions prior to the training of a flow-based generative model. It further uses the calculated SDOT plan as coupling in order to minimize the curvature of the resulting sampling trajectories.

**Strengths:**

The paper is well-written. The motivation is clear. The proposed approach is simple and can be used as a plug-and-play method to improve flow-based models. AlignFlow consistently improves sampling efficiency and convergence over the baselines.

**Weaknesses:**

1. The paper lacks discussion to some prior work that also attempted to improve minibatch-based OT coupling in flow-based generative models (see [1, 2, 4]).
2. The paper would benefit from additional analysis of how scalable the Algorithm 2 is with respect of the data size. Besides this, some additional details on how efficient the OT plan calculations are would be appreciated (Equation 5 and steps 6 and 8 in Algorithm 2).
3. The method assumes the dataset is available as whole prior to the training. However, modern generative models trained at scale sometimes deal with streaming data that is not available before the training. Could the authors provide some discussion on this?
4. While the paper suggests a way to deal with class-conditional training, it is not clear how it would extend to more complex conditioning signals, such as text. [3] propose a solution toward this. Could the authors provide some discussion whether this or similar techniques could be applied to AlignFlow?

[1] Davtyan, Aram, et al. "Faster inference of flow-based generative models via improved data-noise coupling." The Thirteenth International Conference on Learning Representations. 2025.

[2] Zhang, Stephen, et al. "On fitting flow models with large sinkhorn couplings." arXiv preprint arXiv:2506.05526 (2025).

[3] Cheng, Ho Kei, and Alexander Schwing. "The curse of conditions: Analyzing and improving optimal transport for conditional flow-based generation." arXiv preprint arXiv:2503.10636 (2025).

[4] Calvo-Ordonez, Sergio, et al. "Weighted conditional flow matching." arXiv preprint arXiv:2507.22270 (2025).

**Questions:**

1. At the first glance the optimization problem for rebalacing in Equation 12 has a large search space. Could the authors provide more details on how this is implemented in practice?

Some typos:
1. Line 131. The sentence seems to miss a word.
2. Line 208. smaple $\rightarrow$ sample.
3. Line 398. I believe Semi-Discrete Optimal Transport was meant instead of Stochastic Denoising Optimal Transport.

---

> ### Author Response · Authors · 2025-11-21
> **Response  Part 1/2**
>
> We appreciate the reviewer for pointing out the plug-and-play advantage of our method and the empirical improvements in our paper.
>
> Weaknesses:
>
> > 1. The paper lacks discussion to some prior work that also attempted to improve minibatch-based OT coupling in flow-based generative models (see [1, 2, 4]).
>
> We thank the reviewer for providing the missing references; they are added in the revised version with discussion in page 3.
>
> [1] improves upon minibatch OT by computing the map between a pre-sampled noise set and the whole dataset. Nevertheless, it is limited by its ability to handle only a finite number of noise samples.
>
> ~~[2] provides algorithms that scale the Sinkhorn algorithm to large-scale datasets. However, similar to [1], the noise also requires pre-calculation. Moreover, their OT is performed after PCA in a lower-dimensional space, which could potentially hurt performance.~~
>
> [2] provides an approach that scales the Sinkhorn algorithm to large-scale datasets, computing OT couplings between sampled noise and data points in large batches. Moreover, PCA can be used to further speed up Sinkhorn computation, which [2] reports does not sacrifice generation quality.
>
> [4] is similar to minibatch OT, but introduces a weight from Gibbs kernel in the FGM training objective, and can be viewed as a generalization of minibatch OT.
>
>
> [1] Davtyan, Aram, et al. "Faster inference of flow-based generative models via improved data-noise coupling." The Thirteenth International Conference on Learning Representations. 2025.
>
> [2] Zhang, Stephen, et al. "On fitting flow models with large sinkhorn couplings." arXiv preprint arXiv:2506.05526 (2025).
>
> [4] Calvo-Ordonez, Sergio, et al. "Weighted conditional flow matching." arXiv preprint arXiv:2507.22270 (2025).
>
>
> > 2. Scalability of Algorithm 2 w.r.t. the data size.
>
> Thank you for noticing this important part. To the best of our knowledge, the total complexity of computational cost for SDOT map w.r.t. the data size is generally an open problem. [5] shows when being optimized by SGD, the SDOT objective converges in $\mathcal{O}(1/k)$ when $\epsilon>0$ and $\mathcal{O}(1/\sqrt{k})$ when $\epsilon=0$ where $k$ is the number of iterations, and each iteration has computation cost linear in dataset size $|I|$. Algo 2 uses Adam, which is generally faster than SGD.
>
> [5] Taşkesen, B., Shafieezadeh-Abadeh, S. and Kuhn, D., 2023. Semi-discrete optimal transport: Hardness, regularization and numerical solution. Mathematical Programming, 199(1), pp.1033-1106.
>
>
> > 2. Efficiency of the OT plan calculations (Equation 5 and steps 6 and 8 in Algorithm 2).
>
> In all experiments, we set $\epsilon > 0$, corresponding to Step 6 in Algorithm 2. The average runtime for each noise sample in Step 6 is approximately $4.9 \times 10^{-6}$s on CIFAR-10 and $8.1 \times 10^{-7}$s on ImageNet. Additionally, we evaluated the runtime for Step 8; for each noise sample it is approximately $3.8 \times 10^{-6}$s on CIFAR-10 and $6.2 \times 10^{-7}$s on ImageNet.
>
> Evaluating the SDOT map requires scanning the whole dataset, i.e., complexity $\mathcal{O}(|I|)$ with a small constant per sample. It is worth noting that in modern neural networks, the number of parameters is generally much larger than the number of data points, so the forward pass and backward propagation are far more expensive than evaluating the SDOT map. In practice, we found that the total process, including computing and evaluating the SDOT map for hundreds of epochs, increases cost by less than 1% in the ImageNet experiments; this low-cost property scales to large datasets. We have added this discussion in the revised version in Rmk.4 in page 14.
>
>
> > 3. AlignFlow with streaming data
>
> We admit our current framework cannot handle the streaming data, since computing the SDOT plan requires the complete dataset. We appreciate the reviewer for pointing this out, and this is a very interesting future work. Our current idea for achieving this is as follows. When a new data point arrives, we update the SDOT plan. The new dual weight of the new data point is initialized as the dual weight of the nearest neighbor of the new data point in the dataset. Based on the intuition that the dual weight for SDOT plan will only slightly change with only one new data, the update for SDOT plan could possibly be cheap.

---

> ### Author Response · Authors · 2025-11-21
> **Response Part 2/2**
>
> Weaknesses:
>
> > 4. How to extend AlignFlow to more complex conditioning signals, such as text. Can the techniques in [3] be applied?
>
> As the reviewer points out, complex conditioning signals are important and we admit that applying AlignFlow to this task with infinite, complicated signals is challenging. We would provide some initial thoughts for how AlignFlow may be used to address this task.
>
> Suppose the data is given by tuples $(x, y)$, where $x$ is the image and $y$ is the text (or other signals), and the task is to train AlignFlow that generates new $\tilde{x}$ given some new $\tilde{y}$. We can cluster the text $y$ (e.g., via an LLM or extract the text embedding of each text $y$ and apply a traditional clustering method) and assign a label $z=\mbox{clustering}(y)$ to each $y$, making the input data $(x, y, z)$. For example, $y_1=$"a dog is swimming" and $y_2=$"a dog is running" could be clustered into the same cluster. Then, for each cluster, we compute the SDOT map $\varphi|_z$ to map noise to images $x$. Training the flow-based generative model with $v(x|y)$ can be guided by the corresponding SDOT map $\varphi|_z$, where $z$ is the class label for $y$.
>
> Although [3] addresses OT for Flow Matching in conditional and minibatch contexts, its dependence on permutation matrices for aligning samples and labels prevents easy extension to SDOT. Specifically, the continuous transport map inherent to SDOT cannot be used to permute finite data labels in the manner this method requires.
>
> Extending our work to text-to-image generation will be very interesting future work, and we have included this idea in the Future Work section in our revised version in page 10.
>
> [3] Cheng, Ho Kei, and Alexander Schwing. "The curse of conditions: Analyzing and improving optimal transport for conditional flow-based generation." arXiv preprint arXiv:2503.10636 (2025).
>
>
>
> Questions:
> > 1. More details about rebalance in Eq. 12
>
> Thank you for the suggestion of providing more details for Eq. 12. This equation addresses the rebalancing of the data point frequency histogram to a uniform distribution. This is achieved with linear complexity: utilizing the SDOT map, we estimate point frequencies via noise sampling and shift mass from over-represented points to those with lower frequencies. We have included the rebalancing algorithm in the Supplementary Material. It is implemented on CPU without optimization algorithms.
>
> > Typos
>
> Thank you for pointing out the typos. We have fixed them in the revised version and proofread the whole paper.

---

> ### Public Comment · ~marco_cuturi1 · 2025-11-22
> **Additional details on reference 2**
>
> The authors write
>
> > *[2] provides algorithms that scale the Sinkhorn algorithm to large-scale datasets. However, similar to [1], the noise also requires pre-calculation. Moreover, their OT is performed after PCA in a lower-dimensional space, which could potentially hurt performance.*
>
> As one of the authors of that reference, please allow me to correct the assessment above.
>
> First, *all experiments in the main body of our paper are done in full dimensional space*.
>
> Second, The PCA approach outlined in p.5 is provided as an alternative to scale up the computations of Sinkhorn coupling matrices for high-dimensional data. That approach is mentioned at the bottom of p.5, and further explored in Appendix A.4, **Sinkhorn Speedup from PCA**. That approach is only proposed to alleviate computations. Those couplings can be computed in the original $d=12,288$ space for ImageNet 64, but also, in lower dimensions (see Table 3). We see  no decrease in performance in our experiments when coupling noise to data in PCA space down to $k=500$ principal components.

---

> ### Author Response · Authors · 2025-11-23
>
> We thank the author of [2] for the detailed clarification and for taking the time to engage with our work.
>
> We apologize for the mischaracterization regarding the dimensionality of the experiments in [2]. We appreciate you pointing out that the main experiments were conducted in the full-dimensional space and that the PCA approach is provided as an optional speedup. We also noted the results in Table 3 regarding the performance stability in the PCA space.
>
> We have updated our manuscript to reflect this. The relevant section now reads:
>
> "Zhang et al. (2025) provides an approach that scales the Sinkhorn algorithm to large-scale datasets, computing OT couplings between sampled noise and data points in large batches. Moreover, PCA can be used to further speed up Sinkhorn computation, which Zhang et al. (2025) reports does not sacrifice generation quality."
>
> Thank you again for helping us improve the accuracy of our related work section. We are grateful and will acknowledge your help in a de-anonymized version.

---

### Official Review · Reviewer_k6zR · 2025-10-31

**Soundness:** 4
**Presentation:** 3
**Contribution:** 3
**Rating:** 6
**Confidence:** 3

**Summary:**

The paper AlignFlow: Improving Flow-Based Generative Models with Semi-Discrete Optimal Transport proposes a method to enhance flow-based generative models (FGMs) by explicitly aligning noise and data samples through Semi-Discrete Optimal Transport (SDOT). Traditional FGMs sample noise and data independently, leading to curved generative trajectories and high computational cost. AlignFlow introduces a two-stage training process: first computing a deterministic SDOT map that partitions the noise space into Laguerre cells matched to data points, and then training the FGM using these fixed correspondences. This approach bypasses the curse of dimensionality inherent to standard Optimal Transport methods, scales efficiently to large datasets, and adds negligible computational overhead. Experiments on CIFAR-10 and ImageNet show that AlignFlow consistently improves generation quality and convergence speed across multiple state-of-the-art models—including Flow Matching, Shortcut Models, and MeanFlow—while maintaining compatibility as a plug-and-play module for modern generative frameworks.

**Strengths:**

- The paper presents a new and relevant method to pair data and noise samples when training FGMs. The proposed method is similar in spirit to mini batch OT, but instead of computing the coupling between the empirical data distribution and an empirical version of a Gaussian distribution, it computes the coupling between the empirical data distribution and the Gaussian, using Semi-discrete OT.

- The experimental results show an improvement over the training performance of mini batch OT, and the proposed method is faster as well.

**Weaknesses:**

As in minibatch OT, it is unclear how the proposed approach can be adapted to settings like text-to-image, where we have one or few samples per “class”/prompt, as the plan must have Gaussian marginal for each class/prompt, but this is statistically hard to enforce when there are few samples. Yet, most large scale FGM are text-to-“modality” instead of class-based. To my understanding, this limits the applicability of the method. Could the authors comment on this? Have they found a way to apply their method in text-to-“modality” settings?

**Questions:**

The authors emphasize that unlike for minibatch OT, for which the OT plan is subject to the curse of dimensionality, semi-discrete OT bypasses the curse of dimensionality. However, that is with respect to the OT plan between $p_0$ and the empirical distribution $\tilde{p}_1$. However, arguably the OT plan that we want to be close to is the one between $p_0$ and the population distribution $p_1$, and I assume semi-discrete OT is still curse wrt this one. Is that so? If so, what are the advantages of not being cursed for the plan between $p_0$ and the empirical distribution $\tilde{p}_1$?

---

> ### Author Response · Authors · 2025-11-21
>
> We thank the reviewer for confirming the novelty and improvement! For weakness and the questions:
>
> Weaknesses:
> > How to apply AlignFlow to text-to-“modality” settings?
>
> We appreciate the reviewer introducing this interesting direction--the task of text-to-"modality". As the reviewer points out, we admit that the current AlignFlow for this task with infinite, complicated labels can be challenging.
>
> Here we can provide some initial thoughts for how AlignFlow may be used to address this task: suppose the data is given by tuples $(x, y)$, where $x$ is the image (let's take the image modality as an example) and $y$ is the text, and the task is to train AlignFlow that generates new $\tilde{x}$ given some new $\tilde{y}$. We can cluster the text $y$ (e.g., via an LLM or extract the text embedding of each text $y$ and apply a traditional clustering method) and assign a label $z=\mbox{clustering}(y)$ to each $y$, making the input data $(x, y, z)$. For example, $y_1=$"a dog is swimming" and $y_2=$"a dog is running" could be clustered into the same cluster. Then, for each cluster, we compute the SDOT map $\varphi|_z$ to map noise to images $x$. Training the flow-based generative model with $v(x|y)$ can be guided by the corresponding SDOT map $\varphi|_z$, where $z$ is the class label for $y$.
>
>
> Extending our work to text-to-image generation will be very interesting future work, and we have included this idea in the Future Work section in our revised version in page 10.
>
> Questions:
>
> > The OT plan that we want to be close to is the one between the source distribution $p_0$ and the population distribution. I assume semi-discrete OT is still curse wrt this one. Is that so? If so, what are the advantages of not being cursed for the plan between $p_0$ and the empirical distribution?
>
> Thank you for the insightful question. We would respectfully point out that in our paper $p_1$ is for the empirical distribution and $\tilde{p}_1$ is the unknown population distribution. We agree that "semi-discrete OT is still curse w.r.t. this one". The question "what are the advantages of not being cursed for the plan between $p_0$ and the empirical distribution $p_1$?" points directly to the core of our innovation.
>
> Computing the SDOT map between the noise and the empirical distribution is not cursed, which is what we have achieved. The SDOT map determines the optimal coupling between two fully defined distributions (the noise and the empirical data distributions), allowing for a theoretically precise computation that yields straight transport paths. This stands in contrast to sample-based OT methods, which suffer from the curse of dimensionality and produce biased transport plans. Because sample-based OT relies on couplings computed between samples, the transport plan varies with each iteration. Consequently, the expectation of these biased plans over the entire training process fails to maintain a straight trajectory.

---

### Meta-Review · Area_Chair_id9J · 2025-12-25

**Summary:**

This paper proposes a flow matching model training method that utilizes semi-discrete optimal transport (OT) coupling. This is an improvement over standard batched OT because it requires computing only a single dual weight per data point, allowing for linear-time coupling computation relative to the data size. Reviewers generally supported the paper, giving it borderline accept scores. They highlighted the novel, scalable OT-coupling approach and the supporting experimental evidence. Concerns were raised regarding its adaptation to class/text-conditioned generation, the sensitivity of dual weight calculation, and missing data scaling laws. The authors successfully addressed these concerns in the rebuttal, leading the AC to conclude that the paper meets the bar for ICLR acceptance.

**Reviewer Concerns:**

please see above.

**Reviewer Scores:**

please see above.

---

### Decision · Program_Chairs · 2026-01-26

Accept (Poster)